# Beyond the Power of Audio in Native Transmedia Storytelling: Synergies between Fiction and Reality

**Irati Agirreazkuenaga** [1,*] and **Mikel Ayllon** [2]

1   Journalism Department, University of the Basque Country, 48940 Leioa, Basque Country, Spain
2   Piszifaktoria Ideas Lab, 01400 Laudio, Basque Country, Spain; mikelayllon@gmail.com
*   Correspondence: irati.agirreazkuenaga@ehu.eus

**Abstract:** This paper analyses a transmedia universe featuring a web series revolving around interpersonal relations and gender issues, which was then expanded using a musical podcast and a live concert. The project aims to discover production and narrative development strategies that define a native transmedia production, and how it might be expanded to build stories around complex and multidimensional contemporary themes. An analysis model is proposed to identify the fundamental structural characteristics of the transmedia universe. The research is initially powered by a qualitative approach to ethnographic fieldwork by means of participant observation, and subsequently, the analysis of the interrelationship between the elements that make up the transmedia system. The results show the planning behind constructing transmedia storytelling that intertwines fiction and reality to assist in understanding complex and multidimensional topics such as gender identity or intergenerational relations. The conclusions show that the type of transmedia construction being presented and the position of the audio in that universe is motivated by reinforcing the creation of content that is covered in fiction, but with particular relevance in the real world.

**Keywords:** transmedia storytelling; web series; research methods; podcast; public service; gender





## 1. Introduction

Storytelling is basic to all human cultures and becomes our main means of structuring and understanding common and shared experiences (Jenkins 2008). Human beings need to tell stories, and they tell them to themselves to devise a minimally rational and recognisable tale that gives meaning to the world around them. The poet and activist Muriel Rukeyser wrote: "The universe is full of stories, not atoms" (Rukeyser 1994). Stories provide the raw material used by the media to give sense to the real world, often through constructing fictitious worlds. When telling a story using different media or platforms, this becomes transmedia storytelling. Jenkins (2009) describes transmedia storytelling as a process where fiction is dispersed through multiple channels in an attempt to create a unified, coordinated experience, where each medium makes its own contribution to developing the story (Rampazzo 2019). These experiences can evolve from a single platform, and subsequently be developed in other formats. On the other hand, storytelling that is devised and conceived as transmedia from the outset is known as native transmedia storytelling (Sánchez 2013). This type of production demonstrates a central plot with a variety of stories that flow through different devices, giving unity to a "storytelling universe" (Scolari 2013; Couldry 2008). This new way of generating content allows the creative people behind it to reach a wider audience, as new narrative strategies expand among the audience through different access channels (Sakamoto and Nakajima 2015; Amorós-Pons and Comesaña-Comesaña 2016).

The complexity involved in telling stories in this way means that it is more difficult to understand the structure of these experiences. The scope of the stories included in this definition is broad, requiring methodologies to be developed to describe, compare and

analyse different experiences (Javanshir et al. 2020). In this context, this research aims to reveal the structure and elements within the transmedia world and the relationship between them. The case study is a project promoted by a public media corporation, which includes a fictional story (web series) integrated in the real world (concert) and a musical podcast (fiction–reality hybrid featuring interviews between real people and characters from the series). The chosen topic targets a young teenage audience and tackles interpersonal relations, love, sexual orientation and body image, which is along the lines of a popular Norwegian series called Skam. This series, produced by the Norwegian public television channel, follows a group of teenagers through their daily lives and has managed to create a transnational fan community (Sundet and Peteresen 2021; Villén Higueras and Ruiz del Olmo 2020).

This research aims to reveal how audio formats such as podcasts can be used to connect fictional and real worlds. In addition, the study departs from conducting research only on the elements of a native transmedia world and how they are constructed, but also looks at how they might be expanded and their possible advantages when addressing complex issues. Emphasis is put on the storytelling presented on the various platforms, plus the description of the characters and how they move between media given that the characters become a very interesting aspect of creating native transmedia strategies. They are considered central elements both in constructing fictional works and in the relationship that the audience forms with these worlds (Evans 2011). This research will also add importance to the idea that publicly funded projects are used to address minority themes.

## 2. Background

This research analyses the case of the Daniel ǀ a ǀ universe, composed mainly of two main characters: Daniel (a transgender character) and Ipar (in principle, a character built with opposite characteristics to the previous one). From its conception, the universe intends to attract different spectra of young people through products developed on various offline and digital platforms, given that spectators frequently combine them in their daily life to strengthen the realism of the fictional worlds (Dena 2009). The Basque Public Service Media Corporation, known by its acronym EiTB, was the main promotor of the project from an economic point of view. EiTB was founded in 1980 as part of the creation of the Basque Parliament and provides public radio and television services to the Basque Autonomous Community, Navarre and the south of France.

Within the aforementioned corporation, two media platforms were fundamental when building the transmedia universe: the official Gaztea YouTube channel and the official EiTB website. The Basque Public Radio and Television Corporation (EiTB, in charge of economic financing and service provision), Marmoka Films (overseeing the technical part and project administration) and Piszifaktoria Ideas Lab (responsible for the original idea plus development of the universe and the art department) have taken part in the production of this transmedia world. The original version of all of these products is in the Basque language, which have been subsequently subtitled in Spanish and English.

The Daniel ǀ a ǀ universe is made up of the following three elements:

(a)   A web series named Daniel ǀ a ǀ, comprising six episodes broadcast on the website created by EiTB for the series, on the EiTB on-demand service and on the Gaztea YouTube channel (gaztea means 'youth' in the Basque language, and it is run by the EiTB youth radio station, a clear leader in youth entertainment in Basque (CIES 2021); the role of Radio Gaztea should be highlighted as a driving force behind the project).

(b)   A real concert with an audience streamed live by the EiTB and Gaztea YouTube channels, and which is included in two episodes of the series.

(c)   A podcast comprising three episodes (available on EiTB Podkast and other digital audio services, as mentioned below).

The elements of the story are described as follows:

(a)   The web series Daniel ǀ a ǀ, a fictional story; the first episode came out on 26 September 2021.

Comprising six ten-minute episodes, it is available on the Radio Gaztea YouTube channel and on Eitb.eus (Basque Public Radio and TV, accessible on https://www.eitb.eus/daniela/, access date: 26 September 2021). In general terms, the topic of the series revolves around interpersonal relations among young people and different issues associated with this age group. The series has won awards for the best Basque digital series and the best actor for Gabriel Ocina at the Bilbao Seriesland festival (in total, 85 web series were included from 26 countries), and Iholdi Beristain was named best actress at the Rome festival, Digital Media Fest. Thanks to the recognition it received, Daniel|a| also took part in the well-known Die Seriale festival in Giessen, Germany, in June 2023 with three nominations.

The fictional story revolves around two characters, aged 21 years old, Daniel and Ipar. They meet at university studying communication and connect straight away. They both like music, and Ipar uses social media (YouTube and Instagram) to release her own music.

(b)     The concert was held on 23 September, and it featured in the last episode of the series.

Ipar (Iholdi Beristain), one of the main characters in the web series story, gave a concert alongside other musical groups, which was open to the public. The final scenes of the series were recorded there. Radio Gaztea held a prize draw to give away tickets to the concert, and it could be watched live on eitb.eus. Putting on a live concert was important for Ipar, given that she had only released her music on social media until then.

(c)     The Ipar orratza podcast came out on 8 September 2021.

Podcasts are radio creations by individuals or media that form part of their content expansion strategy; these are periodic digital audio publications that are downloaded from the Internet using web syndication and they change how content is structured and sent out, as well as changing the interaction with users (Chomón-Serna 2016).

The podcast, known as Ipar orratza (brújula in Spanish and compass in English), comprises three 25 min episodes starring two fictional characters who present the programme: Ipar from the series and a barman, plus the leaders of real bands who are guests on the programme. This product is classified as 'musical' within the categories created on the EiTB Podkast platform. It is also available on Spotify, iVoox, Apple Podcast and Google Podcasts.

## 3. Materials and Methods

This research followed a qualitative approach (Sanmartín 2003) backed up by ethnographic fieldwork by means of participant observation and by applying our own analysis tool to study the content of each aspect of the transmedia world and how it is all interrelated. On the one hand, data were collected several times in the early production phases, using the participant observation ethnographic analysis (Jerolmack and Khan 2018; Angrosino 2012) to systemically study how each product was created and developed from the content creation section.

With regard to the ethnographic work carried out, the observation period was implemented in three phases between 1 June and 30 September 2021, in which members of the production and the artistic crew were observed. In the first phase, called pre-production and carried out between 1 and 30 June, the creative work between the scriptwriter and the producer was observed, setting out from the beginning the transmedia nature of the project and how this was going to be developed for the different platforms, as well as establishing communication with the public service broadcasting company. In this phase, we also observed how the script was developed, i.e., the conception and development. In the second phase, called production and carried out between 1 and 15 July, the production team was observed with regard to the technical approach to the transmedia project, including the filming, the recording of the podcast and the production of the concert. During these three moments, the artistic side was also observed, especially by focusing on the two main actresses as they played their roles on the different platforms. In the last phase, named post-production, the editing team was observed as they followed the guidelines devised

to finish forging the transmedia universe. A multidisciplinary post-production team was formed so that the whole project could be controlled from the same place.

As seen in previous research on transmedia created by the Basque Public Service Media (Larrondo et al. 2020), it is still difficult to implement wholly transmedia products given the internal resistance, which normally manifests as different perspectives when it comes to understanding the audiovisual world and the value of each medium within the corporation. However, the conception and progress of the Daniel l a l universe is considered successful in this respect, as it could be developed in full despite the difficulties encountered. Furthermore, from the start of this project, when conceiving the audiovisual storytelling, it was decided that it would be developed on different platforms using different types of language; this made it native transmedia.

In the initial phase of the research, participant observation firstly took place during the key periods of creating the story in its different spaces: the web series production (pre-production, shooting and post-production), the podcast episodes (pre-production, recording in the studio, editing and post-production) and the live concert. Thanks to the field diary tool, which originates from ethnographic methods (Paterson and Domingo 2008), data could be collected in situ while the elements of this multidimensional storytelling narrative were constructed. The participant observation method was based on a field diary structured to collect data around the transmedia structure of the universe, the construction of its narrative and the hybridisation of genres in the elements of the storytelling, including the podcast. All of the content was extracted following the various descriptors: central structure, characters involved, genre of the content being worked on, and translocation of real and fictitious characters.

In addition, the proposed analysis matrix (Table 1) was applied, as it is a tool that can be extrapolated for the study of other cases. It considers a series of items based on the narrative characteristics that define the product strategy. The analysis matrix is original and was inspired by work to analyse transmedia media and native transmedia content from Jenkins (2009), Beddows (2012) and Javanshir et al. (2020). The data entered in this matrix were based on the content mining results from the series and the podcast, including all of its episodes. The topic, characters and the genre of the content and how these elements are interlinked were analysed. Furthermore, it was studied how fiction is incorporated into the real world and vice versa, with the intention of detecting the storytelling elements that strengthen the link between reality and fiction.

**Table 1.** Strategies and narrative elements used in the Daniel l a l transmedia universe.

| Strategies and Narrative Elements Being Analysed | Narrative Characteristics Observed |
|---|---|
| 1. Strategy for expansion and interrelationship between the different platforms; connection between narratives and continuity | The web series incorporates a concert and extends into a podcast. One medium tells the story and the others complement it while remaining independent. The music appears as a binding element for the transmedia universe. The live concert offers an immersive space. |
| 2. Real and fictional characters on the various platforms | There is a translocation of characters as they appear on all three platforms. The web series is devised through the eyes of the character of Daniel, and the podcast and the concert through the character of Ipar. |
| 3. Key connection elements between fiction and reality | The podcast on the radio is the connection point for the three media. |
| 4. Seriality | The story is extended in several instalments on the same medium and in multiple media: six chapters in the web series and three in the podcast that are offered as a series. |

## 4. Results

The results of analysing the strategies and the connection between elements in the transmedia universe are given in Table 1 and explained below.

### 4.1. Strategy for Expansion and Interrelations among the Different Platforms

This is a global creation process (native) that conceives the product in its entirety from the origin, so that the content is not adapted to the different media, and the various platforms remain relatively independent from the main narrative, although the conflict that is being demonstrated is the same. A universe is created using the languages belonging to the three supports, and each of them is used to favour or exploit a specific field of the story using initial planning.

The backbone of this project is a fictional story, and the main story is told in the web series. A live event is integrated into the web series, where the character of Ipar sings live for a real audience that is attending a concert in a venue close to Bilbao. The podcast joins the two previous items using the same theme and characters. The web series tells the fictional story by itself and the other two supports complete it, although they are independent. Each entry is independent, so the concert can be enjoyed without watching the series and vice versa.

Among other topics, the web series addresses the topic of interpersonal relations among young people and the topic of transgender identity based on a fictional story, where music takes centre stage. In fact, one of the main characters gives the live concert that is included in the series with real people and characters from the series in the audience. Finally, in the musical podcast, some voices come from fictional characters, such as Lur the barman and Ipar, who lead the interviews, and others are real people such as singers from the bands who are interviewed; however, the main narrative is always maintained. Although the same theme runs throughout the media, the characters' narrative conflict varies according to the platform. In this respect, the characters obtain some autonomy from the basic story and act independently as interaction elements.

The concept of continuity becomes particularly relevant in this case and refers to the elements that give continuity to the macro story. In the case analysed here, three elements contribute to this aspect:

1. The character of Ipar: she co-stars in the series where she plays the role of an undergraduate journalism student; she is the journalist interviewing the singers in the music podcast; and she is the protagonist, the main singer, playing live at the concert.
2. The music: the Basque female singers who make up the soundtrack of the series are interviewed in the podcast (conducted by Ipar), and in the live concert one of these bands plays after Ipar.
3. The target universe of the radio: it appears as an object within the narration of the web series; in the podcast itself, which after all is a variant of radio; and in the concert that was broadcast via streaming by Radio Gaztea.

These three elements provide continuity using the shared spaces that have been devised alongside the sound environment. In addition to the main issue, the music takes centre stage through the series characters' interests, with the live concert and the musical podcast being constructed as a binding element for the transmedia universe. The live concert offers an immersive space, a real-life aspect that becomes part of the fictional life.

### 4.2. Translocation of Characters (Real and Fictional) on Three Platforms

There is translocation of characters across all three platforms: the web series is seen through the eyes of Daniel, and the podcast and the concert, on the other hand, through the character of Ipar. In the web series, one of the main characters, Daniela, who lives on the outskirts of Bilbao and loves urban music (trap, funk, etc.) finds it hard to accept themself and clashes with the people around them. They go abroad in search of another way of life, where they realise the root of their problems and how they intend to live from then on. Consequently, by the end of the first chapter, we see that they have found peace with

themself, and they return home and decide to be known as Daniel. From that moment onwards, we get to know Daniel who embraces their transgender status very naturally.

The other main character, Ipar, is the same age as Daniel. She lives in the city centre and is in her final year at university. She is a hard-working student who gets on very well with her classmates and friends and she is an exceptional communicator. She uses social media expertly to spread her passion for the music that she creates herself. Although she is studying communication, she would like to work in music. The initial episodes reveal that she will shortly give her first concert in front of an audience in a venue in Bilbao, which she sees as a major milestone in her artistic career. She unexpectedly meets Daniel, who turns her life upside down.

One of the binding elements that brings these multidimensional narratives together and inserts them into the conversation is Ipar (a first name that can be male or female). As revealed in previous research (Spinelli 2020; Naldi and Dalla Torre 2022), here too the multidimensionality makes the universe more nuanced and therefore more credible; that is, it makes it more similar to reality and therefore more consistent. In the production of the universe, it can be argued that the transmedia structure gives a fictional story credibility, above all because it fills out multiple dimensions of the characters. These characters appear on the various platforms as they are shaped from different perspectives using different narrative languages, which gives us an almost 360-degree view of each character, that is, a holistic view of each of them. The narrative languages of Ipar's character are seen from a different angle in each space or platform. In the series she expresses herself as Ipar (the character, with her fears and virtues built around the character of Daniel), whereas in the podcast she is a journalist and at the concert a musician; she expresses herself and acts differently. Furthermore, in the case of Daniel, they appear as a passive subject both in the podcast and in the concert, because this character is talked about in these spaces, but the character does not appear.

On the other hand, the podcast uses a classic informative genre via the musical interview and thereby looks realistic. The questions are asked by Ipar based on a pre-arranged script; however, the profile of the actress (Iholdi Beristain) as an audiovisual communication degree student gives the character room to make their own asides during the interviews.

*4.3. Key Binding Elements (Immersion)*

The podcast episodes follow the same structure in four blocks: A brief intro that is used to locate the fictional space, which is followed by a literary section which talks about a book on music and aims to expand the story of the fictional characters from the web series: Ipar and Daniel. Then, the central interview begins, taking up most of the space, where the two fictional characters chat with a guest artist (real character). Finally, one of the fictional characters, Ipar, sings live with the guest artist.

The musical aspect is very important in this universe, as the main character sings and plays the guitar in the web series, and this is taken as the basis to generate unity and connection between the transmedia worlds. The character creation and development, plus linking music to their conflicts, are all essential when constructing the fictional world. Music creates an emotional link with the audience (concert and podcast) and the audience (fans) recognises the characters and takes part in an extension of the main story by attending the concert. The songs represent the topic and the internal conflicts of the characters and can also attract a young audience due to the lyrics and the style of the music.

In the literary section, Lur, the fictional character who accompanies Ipar in the podcast, reads a few lines from a book out loud. This attempts to bring the literary world to the podcast; each episode features a different book, always related to music. One of the episodes highlights a sentence from Emma Goldman's autobiography *Living My Life* (Goldman 1931), "if I can't dance, I don't want to be part of your revolution," which kick-starts a conversation with Ipar on the importance of making some life decisions or giving them a more recreational perspective. At another point, they read another phrase that invites

the audience to think about music, taken from the book *The Descent of Man, and Selection in Relation to Sex* by Charles Darwin (1871): "the suspicion does not appear improbable that the progenitors of man, either the males or females, or both sexes, before they had acquired the power of expressing their mutual love in articulate language, endeavoured to charm each other with musical notes and rhythm". Using this fragment, the characters talk about Ipar's relationship with Daniel that features in the web series.

*4.4. Seriality*

The principle of seriality refers to an amplified version of the traditional series concept, given that significant story fragments are created to divide up this story not only into several instalments on the same medium, but also across multiple media systems. The case being examined meets this principle of seriality through creating the six episodes of a web series and the three episodes of a podcast that are offered in series.

**5. Discussion**

*5.1. The Native Transmedia Ecosystem: Fictional Histories to Understand the Real World*

Native transmedia storytelling implies (re)creating a story and selecting the appropriate platforms to extend it (Costa Sanchez 2013). Consequently, constructing a transmedia product requires a good command of the techniques for each medium: each one requires its own language and its own form of expression to get the best result out of each format (Canalès 2020). In the case being analysed, each medium uses its own language and does so independently while meeting the principle of the "construction of worlds" (Jenkins 2009; Moloney 2011), given that the story of Daniel|a| constructs a universe, including common elements, fundamentally through music. Meanwhile, in non-native transmedia universes, as opposed to making the most of each platform to develop and expand the central story, what we see is repeated content (Robledo-Dioses and Atarama-Rojas 2018). In the universe being analysed, on the other hand, the stories that take place in each space are interconnected while also offering a differentiating aspect to the story.

Transmedia storytelling tends to strengthen the plurality caused by the different characters and stories (Scolari 2013). Consequently, the transmedia extensions can connect with the audiences by contrasting several subjective experiences, such as the perspectives of different fictional characters (Jenkins 2009), and has indeed fostered significant amendments to the production and teaching of media entertainment (Jenkins 2010). In this case, subjectivity is provided in different ways in each medium, given that the web series is created from the gaze of Daniel, and the podcast and the live event (the concert) revolve around Ipar. Furthermore, a sense of immersion, which is understood as the purpose of involving the real audience in the fictional story (Jenkins 2009), is achieved through the concert. In this way, real life becomes fictional life. The concert is not a recording made for the series, that is, the audience in the concert is the audience that has gone to enjoy a live concert, not fictional extras brought in for the series. The concert takes place in a real moment; a moment of reality is recorded. Moreover, in the podcast, a fictional character interacts with real people, interviewing them about their real-life experiences, which interconnects and blurs the boundaries between fiction and reality.

In the case being analysed, it is deduced that a fictional story can be incorporated into the real world without losing the sense of reality and vice versa, where a real situation can form part of a fictional story. This focus creates a strong association between a fictional story and the real world, above all, through the music and the main characters, blurring the border between the two worlds to a certain extent. As a result, the transmedia universe of Daniel|a| creates synergies between reality and fiction, so that, on the one hand, it reinforces the audience's interest in the theme covered in the fiction and, on the other hand, in the musical information being offered (as explained in the following section).

As the use of different narrative strategies helps us to understand the diversity of genres (Bravo 2021), transmedia universes can help to address complex and multidimensional themes that are increasingly present in our societies. This transmedia production

manages to use various perspectives to look at complex themes such as gender transition or interpersonal relations among the teenage-adult population. In this case, we are introduced to the character of Daniel and their concerns, not only directly through the web series story but also through Ipar's eyes, using the podcast and the live event (concert). This all delves deeper into the character and their complex existential situation. Therefore, we can conclude that the transmedia projection offers a more complete strategy to address complex themes, such as gender identity or interpersonal relations, given that each platform shows different sides of the characters and their contexts.

Some scholars have shown that transmedia stories from minority perspectives, such as the feminist point of view, are usually marginal compared to a more conservative focus (Kustritz 2022). In this respect, it is important that it is produced by a public service broadcaster, given that addressing minority issues lies at the centre of Public Service Media (PSM) efforts (Naerland and Dahl 2022). That is to say, making space for cultural diversity constitutes an important part of PSM missions, and this ideal is reflected in both EU and national broadcasting policies.

*5.2. The Importance of Audio: A Podcast That Ties in with the Transmedia Universe*

As opposed to other cases (Miranda-Galbe et al. 2021) where the audio space is the least relevant aspect used to construct a transmedia universe, audio and the world of the radio are key in the universe analysed here. As such, they traverse each of the spaces and are present in all of the products: in the series, in the podcast and in the concert.

The renown of the podcast is seen as a tool to extend the narrative universes that have been created. The episodes of the Ipar orratza podcast combine fiction and journalism genres, using interviews which bring real characters into the fictional world. However, this intertwined relationship goes one-step further, as it not only concerns the characters from the web series, but it also appears through the music heard live in the podcast sung by the real guest artists. Those artists music become the soundtrack for the web series, or it appears as the voice of one of the fictional characters in the podcast that is heard again on the radio in several key scenes in the web series.

The podcast helps to develop the transmedia universe of Daniel|a| by means of bringing together reality and fiction (the web series), and it is the element that links the live concert in a real venue in front of an audience with one of the main characters in the web series. Its production is not limited to the usual interview, but rather it attempts to foster a deeper reflection in the broadcast, establishing a more human connection with cultural traits concerning music and different aspects related to it. Furthermore, the podcast is strengthened through the interest aroused by the appearance of the web series characters, which can attract new audiences who are less used to this medium.

We can therefore conclude that the episodes from this podcast constitute a hybridisation of genres that is unusual in musical podcasts, not only by combining journalism and fiction genres, but also by introducing the literary world through one of the characters who systematically appears in all episodes. At the same time, the content also suggests something more than a musical podcast, with literary texts woven into a discourse between journalism and fiction. Consequently, this study is added to the idea that, after more than a decade of development, podcasting has constructed its own narrative that, far from acting in isolation, creates logical and coherent grammar (García-Marín and Aparici 2018). The Ipar orratza podcast represents an innovative means of production: a musical podcast that transmits the other elements of the transmedia world, which is important to attract youth audiences to the radio. This is possible given the potential that the radio, and particularly podcasts, can have to develop these types of situations (Sullivan et al. 2020). In fact, recent research has demonstrated that young people listen to the radio digitally (Espinosa-Mirabet and Ferrer-Roca 2021), and many turn to YouTube to explore their interests and complement their studies (Pires et al. 2022). Over the last ten years, podcasting has been shaped as a communicative practice that incorporates some of the most outstanding features of digital

media production (García-Marín and Marino 2020) while attempting to attract a youth audience that was not connected to traditional radio (Pedrero-Esteban et al. 2019).

The Daniel | a | project received funding from the Basque Public Media. As several authors have emphasised, it is extremely important that public media finances independent productions to create loyalty among its local audience in terms of consuming media products and producing content to reflect deeper cultural or social meanings (Redvall 2013), creating an alternative counter-flow which is creative and non-commercial (Jensen 2016). However, it was not possible to design an audience participation strategy in this case study as the budget was not large enough, as is true for many other cases mentioned by other authors (Krauß and Stock 2021).

## 6. Conclusions

Audio and the shared sound environment can be key in the transmedia universe. The podcast is an element that expands the narrative universes, creating and binding the story together on various platforms. There is a shared environment created through the music that crosses each of the spaces and is present on all platforms. Therefore, the world built through sound becomes, in this case, not only an additional part of the transmedia universe, but the basis from which the entire potential narrative of the story unfolds. The analysed product, which has been produced and broadcast, shows that sound can be the trigger for the construction of the rest of the languages and platforms within the narrative.

Furthermore, the transmedia strategy helps to create a more complete story on complex and multidimensional themes, such as gender identity or interpersonal relationships, given that each platform can shed light on different sides of the characters and their contexts. Fictional stories are particularly useful tools to improve our experiences and make certain topics easier to understand. Moreover, public funding makes it possible to address minority topics that do not have an entirely commercial purpose.

Finally, within the native transmedia narrative, prior work takes place on structuring the team that focuses on the project design process. Research on this topic can promote an analytical perspective on these processes to design guides that help to consolidate transmedia strategies in line with the requirements of our realities, and can help train future creators. The proposed analysis framework, based on structuring the expansion of the main story and how characters and stories move between platforms, is an effective tool to assess the synergies created between reality and fiction.

Based on the proposed analysis, it could be interesting to analyse in future research whether it is possible to capture the attention of a youth audience that, in principle, has few links with traditional radio through podcasts that share and converge in a variety of media narrative universes and languages.

**Author Contributions:** Conceptualization, I.A. and M.A.; methodology, I.A. and M.A.; formal analysis, I.A.; investigation, I.A. and M.A.; resources, M.A.; data curation, I.A.; writing—original draft preparation, I.A.; writing—review and editing, M.A.; supervision, I.A. and M.A.; funding acquisition, I.A. and M.A. All authors have read and agreed to the published version of the manuscript.

**Funding:** This research was funded by the research group named "Gureiker", a Basque Government con-solidated Research Group (IT 1112) and Piszifaktoria Ideas Lab.

**Institutional Review Board Statement:** Not applicable.

**Informed Consent Statement:** Not applicable.

**Conflicts of Interest:** The authors declare no conflict of interest.

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
