# Peer review of "Beyond the Power of Audio in Native Transmedia Storytelling: Synergies between Fiction and Reality"

_socsci, doi:10.3390/socsci12060344_

Round 1

Reviewer 1 Report

Thank you for the opportunity to review this article. I’d like to firstly congratulate the author on an interesting case study of transmedia storytelling. Learning of new, innovative transmedia projects is always a joy.

Unfortunately, as it is currently submitted, the article required significant reworking.

The main issue with the paper is that the focus and research questions are unclear. The abstract claims that the aim of the project was to find out the production and narrative development strategies that define native transmedia production and to find out how it might be expanded to build stories around complex and multidimensional contemporary themes. When reading through the article however, it jumps in focus from theme, to structure, discussing everything from music and audio, transgender identity to real world and fictional world integration. Even the title is confusing. From the abstract I expected to read about a new analytical model/tool, but this was not produced or discussed adequately. Table 1 is not a tool or a model, so I am unsure how this article presents a new method or design process for transmedia storytelling. The conclusion also begins with the mention of “the transmedia strategy” – is this the same as the model/tool mentioned earlier? I don’t think the claims made on presenting a new tool, model or strategy stands up – at least at the moment. If the author(s) want to keep this focus, they will need to work on demonstrating this more critically. The article’s other topics of discussion were also loosely integrated, and it is not clear how they feature in the tool/model. Is the article claiming that transgender identity is a “multi-dimensional contemporary theme”. Is the article about methods or design? Or is it about structure or themes? It feels like there are 3-4 ideas/questions that are competing to be the focus of the article. The arguments contained within are not substantiated by the content provided, and it required further articulation, explanation, and contextualization within the industrial setting.

The second major issue with the article is that it mentions many contemporary and important scholars working in the field of transmedia, but these citations were used weakly, with no clear connection between previous scholarship and this new research. The arguments made with citations requires significantly greater substantiation from this existing scholarship.

The author(s) make some grand claims without enough evidence or substantiation:

·       When making claims about research outcomes, be careful not to claim too much. For example, on p. 5, lines 230-233 the author(s) claim: “Therefore, we can see that the transmedia strategy might help to raise complex topics such as gender identity or relations between different generations, more completely and accessibly, given that each platform can reveal different sides of the characters and their contexts”. The information provided does not adequately argue this and the argument is therefore not convincing.

·       Similarly, the claims made on p. 7, lines 279-281 (about non-native narratives merely repeating content) – this is not factual. The citation you provided (Robledo, Atarama & Palomino 2017) is not included in your reference list.

·       Again on p. 7, lines 309-311 you claim that “transmedia projection offers a more complete strategy to address complex themes… given that each platform shows different sides of the characters and their contexts”. While you may believe this to be true, this is not substantiated by your project or by scholarly sources. You can rephrase this to “transmedia provides opportunities for further character and theme exploration… etc.).

There are a few other key areas where I feel the article could be strengthened:

·       More background information about the transmedia project, Daniel(a), was needed in the introduction, or in a separate Background section following the introduction to provide the reader with the required context for the project.

·       When discussing previous scholarship, further evidence and discussion is needed. For example, on p.3, line111, Dena 2009 is mentioned, but it is not clear where in Dena’s work she spoke about realism of fictional worlds. Using a page reference here and additional articulation of the connection is important.

·       Some concepts/terminology is used without explanation. E.g “narrative languages” p. 5, line 223. It’s also not clear what is meant by “360-degree view of each character”.

·       Ensure you give your readers the required context and information about the transmedia project. E.g. p. 5, line 242 – what does the character “play”?

·       When you talk about music creating emotional links on p. 5, lines 244-248, you mention that the songs represent the topic an internal character conflicts. But how? Are you talking about lyrics or style of music?

·       The Jenkins quote on p. 7, lines 290-291 needs a page reference in the citation.

·       The sentence on p. 7, lines 312-314 is an incomplete sentence.

·       There are a few typographical errors throughout, which I am sure will be picked up in the copy edit.

The article was most interesting when it was discussing the use of audio and music as narrative tools. How audio formats such as podcasts can be used to connect fictional and real worlds for younger audiences. It was very interesting to read (albeit briefly) about the idea that publicly funded projects are used for minority projects, themes and audiences.

The project and article certainly has merit and would be a welcomed addition to the contemporary discussions on transmedia storytelling, and I look forward to reading the new submission(s).

Author Response

Thank you very much for the comments and observations made. They have been of great help in order to improve the quality of this paper, and I will be willing to make further changes if necessary as it improves the article considerably.

I totally agree on the first point. At this respect, I have changed the approach and, in its present form, it is not really the tool that is being emphasised but rather how it is structured, with special importance given to the audio –particularly when it comes to the main goals–. That is, it underlines the study of the structure to see the role of the different platforms with the different languages. Thus, by restructuring the focus of the objectives, the data content offered could be more concise and focused to provide more articulated, explained and contextualised arguments.

Similarly, the citations of scholars working in the field of transmedia are used in a more interconnected way to give a more holistic sense to their contributions. In addition, a modification has been made to the title following the recommendations.

  • The reference in lines 279-281 is now included in the reference list.
  • The claims made on p. 7 about transmedia projection is now based on the findings of the work.
  • A section entitled Background has been added to provide information on the Daniela project.
  • It has been explained more concretely what we mean in the article by "narrative languages". On the other hand, we add an explanation to the claim about “360-degree view of each character”. In this sense, we mean that a holistic view is offered.
  • We have changed the term "play" on p. 5 to "plays the guitar", to make it easier to understand.
  • On p. 5, it has been added that the songs represent the topic an internal character conflicts due to the lyric and the style of the music.
  • The Jenkins quote on p. 7 has been changed.
  • The incomplete sentence on p. 7 has now been changed.
  • The idea that public service media reinforce minority issues has been further developed through references.

Reviewer 2 Report

This is an interesting case study that I am sure has a lot to offer in terms of new insights and knowledge. It is assumed from reading the paper that there is a large body of data gathered through the research and within this is buried some interesting material. I would prefer a detailed unpacking of just one of the insights rather than a general overview which is what we currently have before us. There are lots of claims made in the paper but rarely are these supported by the research. The author alludes to other examples but assumes that we are already familiar with them. These and other issues need to be addressed before this paper is ready for publication. I am also interested to understand why the author has selected this journal for publication. Given the obvious readership I wonder if the author might take the opportunity to address their interests and concerns e.g where is the social science in this paper?

Line 63 it is more common to undertake a literature review before the methods section, so the reader has a better understanding of the ground from which you are working.

Line 64 yes but please give more information, who were the people you observed, why them, what were they doing etc... there are lots of great website you could take a look at which will give you a step-by-step description of how to structure the methods section. You need to walk your readers through the methods section. Again, don't assume your reader is familiar with your study.

 Lines 71-78 as I haven't read the case study of Basque public radio, I have no idea what you are talking about. Please don't assume your reader is familiar with your research sources. By the end of the paragraph, I am confused as to which project you are now discussing.

Line 88 why a field diary?

Line 91 how and what was mined?

Line 92 where is the matrix why is it not included here.

Line 108 this really needs to open this needs to be at the opening of the section.

Line 134 I am guessing that you don't put URLs in the body text try looking at the author guidelines.

Line 145 'the concert' is this all you are going to say about the concert? It seems important but you tell us very little about how it sits within the transmedia.

Line 163 at this point in time I have no idea of what the story experience is or what is intended. I am told there are some touch points for engagement but not what the engagement is/was. You need to carry your readers with you so they inhabit the same thought space as you do. At this point I have no idea what this is all about or how "STORY" features in your discussion and analysis.

Line 167 this is all very interesting, but I fail to understand how you conceive of a central, unified story experience emerging. The inter relation between the platforms appears to be largely instrumental and I would be more interested in a concern for the experience of the audience who have to navigate and make sense of the experience as a story.

Line 181 how does it do this and how does it do this through a story experience.

Line 191 then some explanation of how the concept of continuity functions in this context would be appreciated.

Line 209 ah... some of the story finally begins to emerge, to have known this early on would have been helpful.

Line 220 please do argue it then... I haven't seen you present any arguments yet.

Line 224 you make a claim here but offer no evidence to support it. how is the story given credibility, how is it shaped by different perspectives, what do you mean by different narrative languages?

Line 230 I don't think we see anything as you have not revealed anything to us yet.

Line 268 the table is perhaps useful but of limited value. It is mostly descriptive in that it outlines some elements of the experience which suggest it might fall into the category of transmedia. But so what? What are you doing here that offers insight or analysis, you really need to do more than establish the experience is a transmedia, your audience will get that from the introduction. What will you tell them that they don't know or should know. What new or interesting insights emerge from your research.

Line 277 how does it meet the principle of the construction of worlds. anyway, surely you are arguing for the construction of 'a' world not worlds.

Line 283 I sort of see the interconnection, but you might want to try and establish this more clearly. Is this the interesting insight? If so, then this should become the main focus of the argument.

Line 291 the concert audience seem quite passive as with the podcast and the web series, I would expect there to be more active production of meaning in truly immersive experiential storytelling. The experience of the audience is important here, yet you have not really referenced them or shown how your participatory observation method has led to the emergence of new insights.

Line 296 where is the evidence from your research data to back up your claims?

Line 315 yet you have not presented any material that would indicate 'impact'?

Line 319 what other cases?

Line 337 I am not sure you have demonstrated anything. Surely music by its very nature is a cultural product?

I very much hope that you can address the issues raised above as there is a need for good scholarship based on case studies in the field of transmedia. I am sure there is plenty within your research that would be of real value to other scholars working in this area.

Author Response

Thank you very much for the comments and observations made. They have been of great help in order to improve the quality of this paper, and I will be willing to make further changes if necessary as it improves the article considerably. Thus, following the suggestions, I have made changes to the suggested pages and lines, from line 64 to line 337.

In line 64, more information has been provided on the observation and the ethnographic method used. This comment was much appreciated as it gave us the opportunity to elaborate on the methodology in a more concise way.

Furthermore, more content on Basque Public Service Media has been added in the background of the article.

In line 91 mined has been changed by extracted, to make it more understandable.

Line 92 was modified so that the matrix is now cited there.

The point made in line 108 has been moved so its now at the opening of the section.

The URLs in the body text has been removed.

In line 191 the concept of continuity has been developed in order to better explain the phenomena occurring in the analysed universe. 

The statements made in line 209 has been explained earlier in the paper now.  

We have offered more arguments now to present the claims made in line 220.

The claims made in line 224 have been further explained so that it could be clearer how is the story given credibility and how is it shaped by different perspectives. We also better explained the meaning and use of different narrative languages here.

When it comes to the idea of the concert in line 291, we did not expressed correctly the main idea of it and it has been now being improved. The line 315 and the paragraph itself has been change to better present the material we have collected. 

The cases in line 319 have been reviewed.

The assertions made on music in line 337 have been removed as it is not a main idea of this article and would take too much space to explain it, and perhaps, it would be confusing.

Reviewer 3 Report

Review - Multi-Dimensional Topics in Native Transmedia Storytelling: Synergies between Fiction and Reality  

The topic of the research is interesting and relevant, its purpose is precisely defined: "The research aims to reveal the production strategies for the storytelling and the actual elements of the native transmedia world, plus how fiction inter-relates with the real world, through storytelling elements and their relationship, and how they are systematically constructed.”

The presentation of the previous works related to the research area is a bit short, this part should definitely be expanded. In the literature review, the authors mention the most important literature, the works of Jenkins, Couldry, but this section must be expanded. After all, much more work from Jenkins could be cited and incorporated into the research.

We learn little about the media consumption habits of the audience (for whom the series was intended). In other words, this context is not presented in the study.

The aims of the research are precise and can be answered with the chosen test method. The methodology of participant observation chosen from qualitative research is good and its presentation as a case study is appropriate. The characteristics of the analysis are adequate:

"1) the strategy 102 for expansion and inter-relation between the platforms; 2) the translocation of the charac 103 ters; 3) the key bonding elements (immersion); and 4) seriality.”

The research results are remarkable, but the analysis part of the study should be expanded and detailed. And the conclusion should be supported more firmly.

The study lists a lot of literature that is not included in the main text:

Angrosino, M. (2012).

Canalès, A. (2020).

Espinosa-Mirabet, S., & Ferrer-Roca, N. (2021).

Mithen, S.J. (2007).

Pedrero-Esteban, L. M., Barrios-Rubio, A., & Medina-Á vila, V. (2019).

Redvall, E. (2013).

Villén Higueras, S.J. & Ruiz del Olmo, F.J. (2020).

Overall, I recommend publishing the study with corrections.

Author Response

Thank you very much for the comments and observations made. They have been of great help in order to improve the quality of this paper, and I will be willing to make further changes if necessary as it improves the article considerably.

In the literature review, the authors have expanded the section mainly citing and contextualizing other works of Jenkins, as recommended.

The paper does not revealed data on media consumption habits of the audiences as it is not the main purpose of the article; the study is focused on the production side of the product.

The aims of the research are precise and can be answered with the chosen test method.

The analysis part of the study have been expanded and detailed.

The references that were missing have been included correctly, following the recommendations.

Round 2

Reviewer 1 Report

The abstract and article still needs reworking to reflect the changes (or lack thereof) made to the focus of the article. The introduction now states: “The research aims to reveal how audio formats such as podcasts can be used to connect fictional and real worlds for younger audiences. For that goal, it would depart from conducting research on the elements of a native transmedia world and how they are constructed.” Yet, this research question is not the focus of the article or at least the article has not been revised enough to answer and focus on this new research question. The abstract still implies that there is a focus on analysing structure, as does the new title. So, it remains confusing as to what this article is doing/saying.

The methods section still suggests that the article: “Consequently, the analysis is based on the following characteristics: 1) the strategy for expansion and inter-relation between the platforms; 2) the translocation of the characters; 3) the key bonding elements (immersion); and 4) seriality. Knowledge on these characteristics leads us to determine a structure for transmedia products and to define the most relevant elements when the intention from the outset is to create a transmedia product (Table 1).”

The conclusion remains unchanged, which again suggests that no substantive changes were made to the article, or that the author has not edited the conclusion to indicate these changes.

Author Response

Thank you very much for this second round and your time to share more comments and observations in order to improve the article. All suggestions have been of great help in the process to enhance the quality of this paper.

Regarding the last observations made, the abstract has been modified taking into account the suggestions, and so has been the title. Now they are both more focus on the goals of the article and they should convey better the main idea that the authors intent to put forward through the paper.  

The introduction and the methods section have been reformed. The main modifications have been made in page 2 and page 4. In the latter, a paragraph has been deleted and substituted by other that explains better the main focus of the article.

The conclusions have also been changed. The second paragraph has been moved and is now in the first place. Moreover, conclusions linked to the world of sound have been added.

Many thanks again for these valuable observations. 

Reviewer 3 Report

After the transformations, the quality of the study became much better. I recommend it for publication.

Author Response

Thank you very much. I also hope it will be published. Your previous comments were very helpful in improving the article. 

Round 3

Reviewer 1 Report

Reviewer feedback has been addressed.